# Synergistic Effect of Motivation for the Elderly and Support for Going Out II: Measures to Induce Elderly Men to Go Out

**DOI:** 10.3390/geriatrics9010021

**Published:** 2024-02-18

**Authors:** Kenta Tsutsui, Shoko Ukita, Masahiro Kondo, Genta Toshima, Naoki Miyazaki, Kengo Nagashima, Mitsuyo Ohmura, Saki Tsuchihashi, Yoshitaka Tsuji, Makoto Katoh, Naomi Aida, Yoshiki Kobayashi, Yui Ohtsu, Yoshihiro Fujita, Shukichi Tanaka, Hiroki Watanabe, Yasushi Naruse, Isao Iizuka, Hiromi Kato, Yumi Mashimo, Takaaki Senbonmatsu

**Affiliations:** 1Department of Cardiology, International Medical Center, Saitama Medical University, Saitama 350-1298, Japan; knt22e@gmail.com; 2Biostatistics Unit, Clinical and Translational Research Center, Keio University Hospital, Tokyo 160-8582, Japan; s-ukita@keio.jp (S.U.); m.kondo1042@keio.jp (M.K.); g.toshima@keio.jp (G.T.); nmiyazaki@keio.jp (N.M.); nshi@keio.jp (K.N.); 3Department of Innovative Biomarker Development, Institute for Advanced Medical Sciences, Nippon Medical School, Tokyo 113-0031, Japan; m-ohmura@nms.ac.jp; 4Department of Radiology, Saitama Medical University, Saitama 350-0495, Japan; saki_07048@yahoo.co.jp; 5Division of General Education, Faculty of Health and Medical Care, Department of General Surgery, Saitama Medical University, Saitama 350-1241, Japan; ytsuji@saitama-med.ac.jp; 6Research Administration Center, Saitama Medical University, Saitama 350-0495, Japan; m_katou@saitama-med.ac.jp; 7Kobayashi Hospital, Saitama 358-0014, Japan; n.aida@ikkoukai.com (N.A.); yoshikobayashi@ikkoukai.com (Y.K.); 8Graduate School of Humanities and Social Sciences, Saitama University, Saitama 338-8570, Japan; ohtsu@mail.saitama-u.ac.jp; 9Mobility Service Department, Koga Software Company, Tokyo 110-0005, Japan; y_fujita@kogasoftware.com; 10Advanced ICT Research Institute, National Institute of Information and Communications Technology, Hyogo 651-2492, Japan; tanakas@nict.go.jp; 11Center for Information and Neural Networks, National Institute of Information and Communications Technology, Osaka University, Kobe 651-2492, Japan; h-watanabe@nict.go.jp (H.W.); y_naruse@nict.go.jp (Y.N.); 12Business Promotion Department Aisin Co., Ltd., Kariya 448-8650, Japan; isao.iizuka@aisin.co.jp (I.I.); kato.23.hiromi@jp.nokgrp.com (H.K.); 13Department of Community Medicine, International Medical Center, Saitama Medical University, Saitama 350-1298, Japan; ymashimo@saitama-med.ac.jp

**Keywords:** gamification, elderly men, encourage going out, Choisoko system

## Abstract

Background: The second demonstration experiment of supporting elderly people going out with the Choisoko system was conducted. The first study showed that for women, friends, shopping, convenience, and events are factors that have the potential to be effective motivational factors for encouraging these women to go out. On the other hand, these factors did not lead to any behavioral change in men. Since there are approximately 15 million men over the age of 65 in Japan, behavioral changes in the entire elderly population will not occur without guidance for elderly men to go out. Methods: Sixteen elderly men and forty-seven elderly women participated. Interestingly, men are far more passionate about games than women. Therefore, we hypothesized that a preference for games could be a hint as to how we might encourage older men to go out. Then, a second demonstration experiment was conducted, and we analyzed the relationship between six game preferences and the frequency of going out. Results: Among gaming preferences, men with gaming preferences such as Philanthropists, Achievers, and Free Spirits showed a tendency to go out. Conclusions: These stimuli may have the potential to be factors that may encourage elderly men to go out.

## 1. Introduction

The total population of Japan is 124.95 million as of 1 October 2022 [1]. The population aged 65 and over is 36.24 million, and the ratio of the population to the total population (aging rate) is 29.0%, the highest rate of aging in the world [1]. In 1950, the proportion of elderly people was less than 5% of the total population, but in 1970 it exceeded 7%, and in 1994 it exceeded 14%. The aging rate has continued to increase, reaching 29.0% as of 1 October 2022. The male-to-female ratio of the population aged ≥65 years in Japan is approximately 3:4, and there are approximately 15 million men over the age of 65 [1].

In addition, the total population in Japan is declining because the long-term mortality rate exceeds the birth rate. By 2056, the population is estimated to fall to between 100 million and 99.65 million [2]. Concomitant with this total population decline, the aging rate is expected to continue to rise owing to an increase in the number of people aged 65 years or over. There were 8.06 million births in Japan from 1947 to 1949, a period that produced the so-called “baby boom generation” [3]. In the second baby boom (from 1971 to 1974), there were over 8 million births, more than in the first baby boom generation [3]. Individuals born in these periods comprise a substantial stratum of the demographic structure of Japan. Therefore, the population of individuals aged 65 years or over will continue to increase until approximately 2043 [4].

This pattern of an increase in the elderly population, together with population decline, has been called the “future of the past” and is a pattern that characterizes current Japanese society. Public transportation is extremely well-developed in large metropolises like Tokyo and Osaka, so being unable to hold a driver’s license is not a problem for many urban elderly people [5]. However, only 20% of Japan is composed of urban areas; the other 80% comprises suburban and rural areas. Owing to a decline in local transportation because of population decline, the social infrastructure still relies on private cars [6]. Although the overall number of traffic accidents is declining, the number of car accidents involving elderly drivers is becoming more noticeable [7,8]. For this reason, people in suburban and rural areas who are too old to drive experience difficulties maintaining a full social life [9]. In addition, opportunities for elderly people to go out and receive care have fallen sharply since the COVID-19 pandemic, leading to a rise in the number of elderly people who stay at home and an increase in frailty in elderly people [10].

In 2016, the number of driving license holders over the age of 75 years was approximately 5.13 million, which is approximately one-third of the population in Japan aged over 75 years, an increase of approximately 350,000 (7.3%) from the previous year [11]. Moreover, this number is estimated to increase. More than 5 million people over the age of 75 years in Japan hold a driver’s license, and about two-thirds of them are men. Therefore, it seems that many elderly people consider a driving license necessary and that many are reluctant to relinquish their license as they get older [12]. Stopping driving also increases the risk of needing nursing care. The Japanese National Center for Geriatrics and Gerontology surveyed approximately 3500 elderly people aged 65 years or over about the relationship between driving status and certification for long-term care. The results showed that elderly people who stopped driving were approximately eight times more likely to need nursing care than those who continued to drive [13]. Under the Japanese long-term care insurance system, the level of long-term care required by elderly people is determined. Elderly people who struggle to live alone can receive nursing care support through this system. Elderly people who relinquish their driver’s licenses are often forced to give up their hobbies, social circles, family farms, and other activities, increasing their level of confinement. Because of this problem, a scheme was established that leads to a life dependent on nursing care [14]. Healthy life expectancy in Japan is defined as the length of time a person can live independently without relying on daily, continuous medical care or nursing care [1]. In 2019, the average healthy life expectancy for men was 72.68 years, and that for women was 75.38 years, with average life expectancies of 81.41 years and 87.45 years, respectively [1]. Therefore, most people will at some time have to cope with a limited period of poor health. It is important to identify individuals who may require long-term care simply because they no longer have a driver’s license. The loss of purpose and enjoyment in life because of the relinquishment of a driver’s license may induce mental health problems and lead to the need for nursing care. For many people, poor mental health may characterize the end of their period of healthy life expectancy [14].

For this reason, many local governments operate on-demand transportation, such as taxis and community buses, even after returning driver’s licenses and are working to support opportunities to go out and curb the promotion of frailty. However, the participation and retention rates of the elderly are surprisingly low. The convenience and degree of freedom are not as high as those of a private car, and there is a possibility that the system has not yet reached the level of increasing the motivation of the elderly to “want to go out” and “want to use it”. In other words, it is necessary to identify motivation factors that induce them to go out more than privately owned cars. In 2021, with support from the Ministry of Economy, as part of joint research by 13 parties, including Iruma City and Aisin Co., Ltd., we conducted a social demonstration experiment in which the Choisoko going out support system incorporated elements that motivated people to go out in daily life, such as shopping for daily necessities and participating in events. The Choisoko system, developed by Aisin Co., Ltd., is an advanced outing support service utilized by numerous local governments in Japan, boasting a high continuation rate among users [15]. It operates similarly to a taxi service, allowing participants to reserve rides via the Choisoko Operating Center. However, it stands out for its optimized approach, coordinating shared rides if multiple reservations coincide. Consequently, the Choisoko system offers a cost-effective alternative to traditional taxis, mitigating financial strain on users. Many municipalities operate the Choisoko system, which collects a reasonable fare, but this demonstration experiment was conducted free of charge. We assessed the user characteristics and experience and found a marked gender gap in the Choisoko system use: For elderly women, friends, shopping, convenience, and participation in events were considered to be factors with the potential to be effective motivating factors in guiding them to go out. They accepted this system, increased the number of times they went out, increased the number of walks, and showed a rehabilitation effect. On the other hand, men did not accept the system at all and did not respond at all to friends and shopping as factors that encouraged them to go out.

Japan has entered a super-aging society, with an increasing number of elderly people living alone or in households consisting only of elderly couples. The prevalence of frail elderly individuals is also on the rise. The frequency of going out among elderly people is considered one of the comprehensive health indicators [16]. With approximately 15 million men over the age of 65, it is crucial to encourage not only elderly women but also elderly men to go out to bring about behavioral changes among the elderly as a whole. Furuta et al. report that older men’s tendency to go out is significantly lower than that of women, and the risk of being below average in their “want to go out’’ is 1.7 times lower for men than for women [17]. On the other hand, there are no studies that show the factors that motivate elderly men to go out. An interesting study shows that there is a huge disparity between men and women in gaming [18,19]. It is clear that men are significantly more interested in games. In a survey of elementary school to high school students, Tobe et al. report that gaming addiction is higher among boys, indicating gender differences in gaming behavior [20]. The system and commercialization project for preventing nursing care through the use of e-sports shows that older men have more gaming experience [21]. In fact, during the first demonstration experiment, a man asked to go to a game center. Based on these findings, we hypothesized that as men are more emotionally attached to games, the tendency to play games may help to encourage elderly men to go out [18,19]. That is, the main research objective of this demonstration experiment was to determine whether elderly men’s gaming preferences could be linked to factors that induce them to go out. In 2022, in accordance with this hypothesis, we conducted a demonstration experiment with support from the Ministry of Economy, Trade and Industry. We describe here the results of the experiment.

## 2. Material and Methods

### 2.1. Subjects

We conducted an observational study of elderly people living in the Miyadera area of Iruma City, Saitama, Japan. The inclusion criteria were used to select people aged 70 years or older with assistance requirements of level 1 or 2 or long-term care requirements of level 1, according to the Long-Term Care Insurance Act in Japan, which indicated that they could walk around their homes. We thoroughly explained the Choisoko system to all participants and made them understand that it allows them to shop and go out easily. The exclusion criteria for participation were as follows: I, elderly people who could not walk autonomously; II, elderly people with severe illnesses who had difficulty walking or who were considered better off not walking; III, elderly people who had difficulty understanding the purpose of the research; and IV, elderly people who did not provide informed consent (Figure 1).

### 2.2. Procedures

All procedures in this study were approved by the ethical committee of Saitama Medical University (authorization number: Dai 2022-012). Informed consent was obtained from all participants. All measurements were performed at two points before the start of the rehabilitation effect of going out with the Choisoko system and 3 months after the start of the rehabilitation effect of going out with the Choisoko system.

### 2.3. Protocol of Transportation and Rehabilitation Effect by Going Out

Evaluation of participants’ physical function was conducted using the Functional Independence Measure (FIM), a basic checklist, grip strength, five times sit-to-stand test, 6 m walking time, and walking speed. The FIM is an activities of daily living (ADL) evaluation method developed by Granger et al. in 1983. The FIM can be used to evaluate the burden of long-term care, is considered to be a reliable and valid method, and is widely used in the field of rehabilitation. The FIM includes 13 items that assess “exercise ADL”, such as meals and movement, and five items that assess “cognitive ADL”. The basic checklist was adapted for elderly people aged 65 years or over to check for physical deterioration. This checklist is a tool used to prevent deterioration in the condition of older people by identifying individuals who exhibit a decline in living function at an early stage and implementing approaches for the prevention of the need for long-term care. The checklist consists of 25 questions. Other physical function evaluation methods were used, including grip strength, five times sit-to-stand test, 6 m walking time, and walking speed. All measurements were performed at two points: before the start of the rehabilitation intervention using the Choisoko system and 3 months after the start of the rehabilitation intervention using the Choisoko system.

### 2.4. Laboratory and Muscle Mass Measurements

Since the subjects were elderly and this is a continuation of the first demonstration experiment, just in case, we measured laboratory parameters and muscle mass to avoid any problems during the demonstration experiment and to elucidate the causes of changes in physical function. Height (cm) and weight (kg) were measured to calculate participants’ body mass index. Body fat percentage and muscle mass were calculated using a body composition analyzer (InBody770, manufactured by InBody Japan Co., Ltd., Koto-ku, Tokyo 136-0071, Japan). In addition, maintaining good posture is particularly important for elderly people, and decreased psoas muscle function may increase the risk of falls, so we measured the muscle mass. Measurements of psoas muscle volume were performed using computed tomography images taken at the L3–L5 levels by a radiologist. Segmentation was manually performed using free software called itk-SNAP, which is a free, open-source, multi-platform software (itk-SNAP: http://www.itksnap.org/pmwiki/pmwiki.php (accessed on 18 December 2023)), and target areas were extracted. Using blood samples from all participants, albumin, aspartate aminotransferase/alanine aminotransferase, blood urea nitrogen, creatinine for the calculation of estimated glomerular filtration rate, leukocytes, erythrocytes, hemoglobin, hematocrit, mean corpuscular volume, mean corpuscular hemoglobin, mean corpuscular hemoglobin concentration, and platelets were measured. All measurements were performed at two points before the start of the rehabilitation effect of going out with the Choisoko system and 3 months after the start of the rehabilitation effect of going out with the Choisoko system. 

### 2.5. Preliminary Questionnaire on Game Preference 

We used a preliminary questionnaire to assess game preferences of all participants. This questionnaire was modified for this research from an existing questionnaire regarding gaming preferences [22]. Marczewski et al. developed the gamification user type Hexad framework based on research on human motivation, game player types, and real-world design experience [22]. The Hexad framework model is intended to cover a wide range of highly gaming-oriented systems, and we believe that this model is potentially suitable for personalizing gaming-oriented systems [18]. Game preferences were classified into the following six types (Figure 2). Socializer indicates the type that is motivated by relationships. Philanthropist indicates the type is motivated by charitable causes. Disruptor indicates the type that is motivated by change and disruption. Free Spirit indicates the type that is motivated by autonomy and self-expression. Achiever indicates the type that is motivated by mastery. And player indicates the type is motivated by obtaining rewards. Each type consisted of 4 questions (total 24 questions), making the content easy for the elderly to understand. Each question was evaluated on a 7-point scale, with 1 point indicating the highest level of interest in the game performance and 7 points indicating no interest in it. Statistical analysis was performed under these conditions.

### 2.6. Statistical Analysis

The analysis population was defined as participants for whom both data on the number of times they used the Choisoko system and six gaming preferences were measured. All analyses were conducted by sex. Scatter plots and Pearson’s correlation coefficients were drawn and calculated for all combinations of total gaming scores to evaluate the relationship between the scores of each gaming preference.

Four types of analyses were conducted to explore the association between the frequency of use of the Choisoko system and game preferences: (1) Scatter plots and Spearman’s correlation coefficients between the number of uses of the Choisoko system and each total score of game preference were created. (2) Dividing the low- and high-frequency groups of the Choisoko system use, the frequency of use of the Choisoko system and six gaming preferences were compared. The low- and high-frequency groups were defined by the median for each sex. (3) Dividing the no, few, and many use groups of the Choisoko system, the frequency of use of the Choisoko system and six gaming preferences were also compared. Means and standard deviations of each game preference score were calculated, and box-and-whisker diagrams were used to confirm each score’s distribution. (4) A decision tree analysis was performed. A tree was created to classify high-frequency users and low-frequency users. Based on the median value, high-frequent users were defined as those with four or more uses, while low-frequent users were defined as those with less than four uses. The tree was created by CART (Classification and Regression Tree), and variables included age, sex, and total scores for each of the six game preferences. The detailed parameters were as follows. The maximum tree depth was 5, the minimum number of data in the node to be attempted to split was 10, and the minimum number of data in the leaf node was 3.

All 95% confidence intervals (CIs) were two-sided, and all 95% CIs in mean differences were calculated using Satterthwaite method. Statistical analyses were performed using SAS software version 9.4 (SAS Institute, Inc., Cary, NC, USA) and R version 4.2.3 (R Foundation for Statistical Computing, Vienna, Austria) with the “rpart” package [23].

## 3. Results

### 3.1. Baseline Characteristics

The baseline characteristics of the male and female study participants are shown in Table 1. We obtained consent from 18 men and 47 women as subjects; however, 2 men did not participate, leaving 16 men and 47 women as the research subjects. The mean age of male subjects was 83.3 years, and the mean age was 82.1 years for females; the proportion of males was 25.4%, and the proportion of females was 74.6%. Significant differences were observed between men and women in height, weight, left and right grip strength, and muscle mass (Table 1). Appendix A compares the amount of change in each measurement value before and after using the Choisoko system for men and women.

The median number of times the Choisoko system was used was two times for men and six times for women (Table 2). Similar to the previous demonstration experiment, women appear to have a greater affinity for such systems.

### 3.2. FIM Measurement

Table 3 shows the FIM scores of men and women after using the Choisoko system. Of the three scores, the maximum difference in point estimates between men and women is 0.88 (95% CI: −4.90 to 6.65). All scores were not much different between men and women. 

### 3.3. Elderly Game Preferences

Table 2 also shows the average and median values of game preferences of the subjects by gender. The average game preference for the male group was lower than that for the female group (Figure 3). As mentioned above in Section 2, lower scores indicate more interest in the game. The difference between men and women was particularly large for the players’ R-total score, which indicates a gamer who likes athletes, actors, and acting and tends to prefer sports games and role-playing games. Next, we analyzed the correlations between the total values for each game preference using the Pearson product-moment correlation coefficient for each group (Figure 4a,b and Table 4). An absolute correlation coefficient of 0.4 or higher indicated a combination of (P-total, F-total), (A-total, F-total), (R-total, P-total) for men, and a combination of (P-total, S-total), (P-total, F-total), (A-total, P-total), (A-total, S-total), (A-total, F-total) for women. The combinations of (P-total, F-total) and (A-total, F-total) were common to both men and women. This is because, in many cases, the game performances overlap, with not just one game performance. For example, a person with a Philanthropist-like gaming preference also has a Free Spirit-like gaming preference. To investigate the relationship between each game preference type and the Choisoko system usage in each group, we divided participants into two groups according to the Choisoko system usage: a group whose Choisoko system usage was equal to or greater than the median value and a group whose Choisoko system usage was below the median value. Confidence intervals based on *t*-distribution were used to evaluate whether there was a difference between the mean values of the two groups (Table 5). In the male group, the maximum mean difference in F-total was 2.54 (95% CI: −0.58, 5.66), suggesting that preference for this type is most closely related to the number of times the Choisoko system was used. In addition, the total game preference values, except for the D-total, were smaller in the group that used the Choisoko system more. The D-total scores showed the opposite trend (Table 5). In the female group, the maximum mean difference in S-total was 1.69 (95% CI: −0.72, 4.11). Regarding the P-total, S-total, and F-total scores, the total game preference value was smaller in the group that used the Choisoko system more often. The A-total, D-total, and R-total scores showed the opposite trend. The F-total scores, which showed the maximum mean difference of 2.54 (95% CI: −0.58, 5.66) in the male group, had a mean difference of 1.17 (95% CI: −1.18, 3.51) in the female group (Table 5).

The scatter plot matrix showed that there is a tendency for a negative correlation between not using the Choisoko system and F-total in the male group, and P-total, S-total, and F-total scores tended to be negatively correlated with not using the Choisoko system in the female group (Figure 5a,b). Furthermore, regarding the correlation between the number of times the Choisoko system was used and each type of game preference, no game preference showed a strong correlation of 0.4 or higher, but the F-total for the male group was −0.37, and for the female group, it was −0.18 (Table 6). Next, for both sexes, we divided participants into three groups according to the number of times the Choisoko system was used (zero times, one to four times, and five times or more) and then evaluated the number, the average value, and the standard deviation of each game preference using a boxplot (Table 7 and Figure 6a,b). Regarding F-total for men, there was a difference in the average value between the group that used the Choisoko system zero times and the other groups. Furthermore, for the R-total scores, the average and median of the total scores decreased in order of the number of times the Choisoko system was used. The opposite pattern was found for the D-total value; the average and median of the total scores increased in the order of the number of times the Choisoko system was used. However, for women, the only differences observed were in the average value and distribution between the group that used the Choisoko system zero times and the other groups, and no differences regarding gaming preferences were observed.

Finally, we conducted a decision tree analysis to calculate the combinations that were most likely to be associated with the six game preferences and Choisoko system usage (Figure 7). Decision tree analysis is a method that uses a tree structure to identify explanatory variables that affect the objective variable. In the low user group, 72.4% (21/29) were correctly distinguished, and in the high user group, 91.2% (31/34) were correctly distinguished. Overall, 82.5% (52/63) were correctly classified by the tree. For both men and women, participants with a P-total score of 13 or higher used the Choisoko system less frequently. Regarding the male group, participants with a P-total score of 12 or less and an A-total score of 9 or less were more likely to be frequent Choisoko system users.

## 4. Discussion

We conducted a second demonstration experiment to explore factors that induce men to go out. Similar to the first experiment, the FIM scores of participants remained stable during the implementation period, and the median number of times that female participants used the Choisoko system was six, and that for men was two, indicating that the Choisoko system is an effective way of encouraging elderly women to go out. Therefore, in the second demonstration experiment, we investigated the factors that indicate strong motivation among men. One of them is gaming preferences. Previous research reports significant differences in gaming preferences between men and women [24]. The study defines so-called gamers as people who play games for more than 30 min a week, a pattern that characterizes 25% of female gamers and 55% of male gamers. Furthermore, it was found that women are more likely to prefer games that have a free-to-play business model, whereas men tend to like games even if they have to spend money on them. Men and women also show substantial differences in their game genre preferences, with men preferring games that involve action, shooting, adventure, racing, combat, and sports. In other words, we can interpret that men have a strong attachment to games and entertainment and are overwhelmingly more likely to become absorbed in them. As a matter of consideration, during the first demonstration experiment, one elderly man requested that he would like to be picked up and dropped off at an amusement arcade. The request was considered, but in the end, transportation to the amusement arcade was postponed. One method of applying game techniques to areas unrelated to games is called “gamification”. We hypothesized that gaming preferences could be used to encourage elderly men to go out. Thus, men have a strong preference for games, and if their preference for games matches their motivation for going out, then if a system is put in place that stimulates their preference for games, they will be encouraged to go out, and they will go out more frequently. We classified gaming preferences into six types and examined the relationship between the number of times the Choisoko system was used and gaming preferences. Figure 2 shows the six types of game preferences. Based on these gaming preferences, we created a questionnaire with four questions for each item (a total of 24 questions) that was translated into Japanese and is easy to understand for elderly subjects. Statistical analysis was performed on the results using a seven-level evaluation.

Interestingly, the average gaming preference for male participants was lower than the average for female participants. The lower the score in the questionnaire, the higher the gaming preference. This result supports the previous differences between men and women regarding gaming preferences. In addition, combinations that showed a strong correlation (0.4 or more) in the analysis of correlations between the total values for each game preference were (P-total, F-total), (A-total, F-total), (R-total, *p*-total) for men, and (P-total, S-total), (P-total, F-total), (A-total, P-total), (A-total, S-total), (A-total, F-total) for women. (P-total, F-total) and (A-total, F-total) showed a strong correlation (0.4 or higher) for both men and women. Individuals are rarely motivated by just one of these gaming preference types; most are always motivated, to some extent, by every type. In a study by Tondello et al., (P-total, F-total) and (A-total, F-total) also showed a high correlation [18]. The participants in this study were elderly people in Japan, whereas in Tondello et al.’s study, the participants were students at computer science schools outside of Japan. It is interesting that similar trends were found in the two studies despite the considerable differences between the participants; this may indicate the reliability of the Hexad framework. As the main purpose of our second demonstration experiment was to explore factors that induce elderly men to go out, we used multifaceted and exploratory statistical analysis to examine the relationship between the number of times the Choisoko system was used and gaming preferences. We divided male participants into two groups based on the median number of times they used the Choisoko system and examined the relationship between the total number of game preferences and the number of times participants used the Choisoko system. F-total showed the maximum mean difference of 2.54, indicating that it was most likely related to the number of times the Choisoko system was used. In addition, participants in the group who used the Choisoko system a lot had lower total game preference values (except the D-total of game preference). This indicates that people who are more interested in games tend to go out. The D-total showed the opposite pattern. The Spearman rank correlation coefficients showed that F-total scores did not reach an absolute value of 0.4 or higher, but the correlation between F-total scores and the number of times the Choisoko system was used was −0.37. In addition, there was a difference in the mean and median values of F-total scores between the group who did not use the Choisoko system and the other groups, and a negative trend was observed in the scatter plot for the frequency of the Choisoko system use.

Incidentally, for female participants, S-total scores had the maximum mean difference of 1.69. For male participants, the maximum mean difference was for F-total scores, and the highest correlation with the number of times the Choisoko system was used was for F-total scores (−0.18); this correlation was less significant for men. This result clearly demonstrates that men and women differ in their preference for games.

We extended the exploratory statistical analysis and used a decision tree to calculate the combinations that were most likely to be associated with the six types of gaming preferences and the Choisoko system usage. For both men and women, participants who scored 13 or higher on the P-total scores, indicating that they did not fit into that preference, were likely to use the Choisoko system less frequently. Among male participants, those with P-total scores of 12 or less and A-total scores of 9 or less tended to use the Choisoko system more frequently. Although some test statistics, such as Spearman’s rank correlation coefficient, were not significant, F-total scores showed a tendency to be associated with the number of times the Choisoko system was used. In contrast, the decision tree analysis showed a tendency for P-total and A-total scores to be correlated with the number of times the Choisoko system was used. In the scatter diagram for each combination of game preference total values, P-total and A-total scores showed a strong correlation with F-total scores. Of the gaming preference types, Philanthropists are motivated by philanthropy, Achievers are motivated by mastery, and Free Spirits are motivated by autonomy and self-expression. These game preference types could help to inform ideas to encourage outdoor activities. For example, the development of a business that involves picking up trash from rivers and mountains would combine these three gaming interests. This type of activity is highly philanthropic, and individuals with such expertise could pass on their expertise to others. This type of activity would satisfy the game preference of Achievers and Free Spirits. Even older men may find more opportunities to go out. The effectiveness of personalization according to the personality traits of participants was demonstrated in applications in user interface design; persuasion technology, which is broadly defined as technologies designed to change the attitudes and behaviors of subjects through persuasion and social influence without necessarily forcing them; and game production [25,26,27,28,29]. Therefore, a system with personalized outing guidance through gamification may be a more effective factor if it is adapted to personality traits and player types. In a study by Tondello et al., the total number of subjects with P-type, A-type, and F-type personalities exceeded 70% of the total, and elderly participants with these characteristics may also constitute a major population [18]. However, the number of elderly men who participated in this demonstration experiment was 16, which is the biggest limitation of this research. Therefore, we made full use of exploratory statistical analysis to obtain results, but although we were unable to find a statistically significant difference, we were able to identify a trend. Additional studies are needed that include larger samples and large-scale surveys to confirm the demographic range of the target participants for each gaming preference. We believe that identification of the P-total, A-total, and F-total gaming preferences could help to encourage elderly men to go out.

## 5. Conclusions

A second demonstration experiment was conducted with the main purpose of exploring factors that induce men to go out. The present study showed some differences between elderly men and women among the six gaming preferences, and men with gaming preferences such as Philanthropists, Achievers, and Free Spirits were more likely to go out. However, this study was limited in generalizability and precision: the limited geographic area, limited target population, as described in Section 2.1, and small sample size. Therefore, more large-scale questionnaire surveys on various target populations are needed to confirm these findings, and they are currently in the planning stage.

## Figures and Tables

**Figure 1 geriatrics-09-00021-f001:**
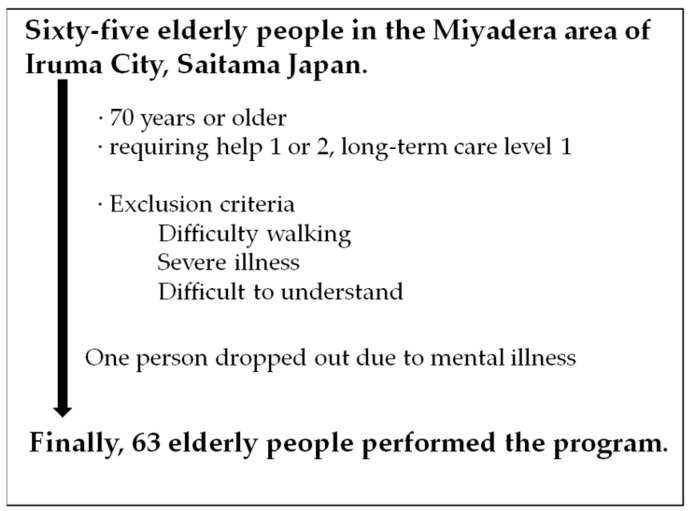
Eligibility of subject person.

**Figure 2 geriatrics-09-00021-f002:**
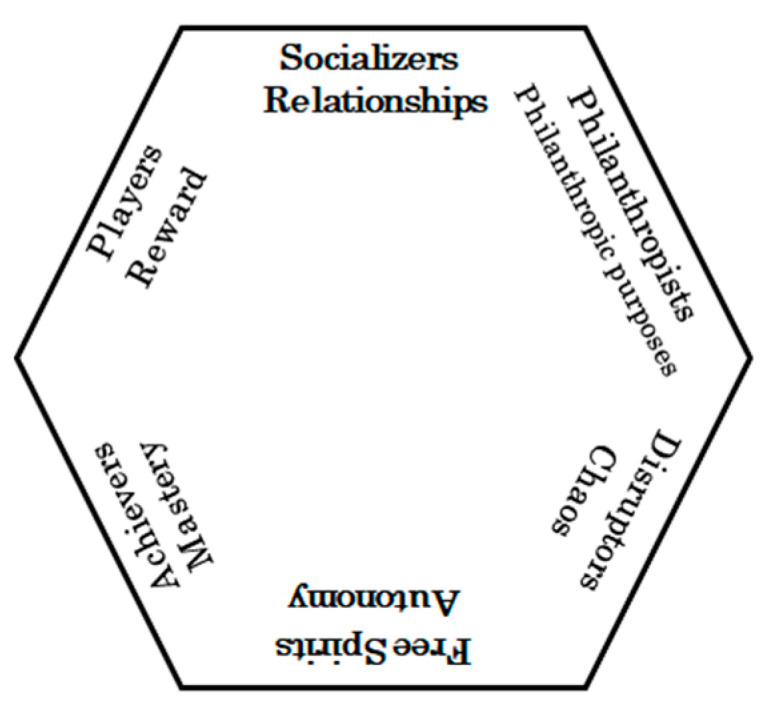
Six types of game preferences. This is a quote from reference [18], © Andrzej Marczewski 2016 (CC BY-NC-ND).

**Figure 3 geriatrics-09-00021-f003:**
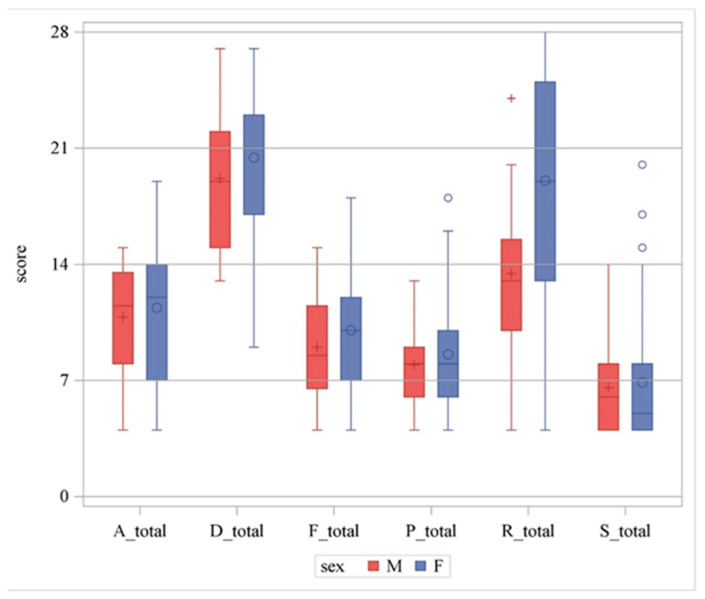
Boxplot for distribution of total of all game preferences. Red bar indicates male. Blue bar indicates female. A_total indicates Philanthropists total score. R_total indicates Players total score. S_total indicates Socializers total score.

**Figure 4 geriatrics-09-00021-f004:**
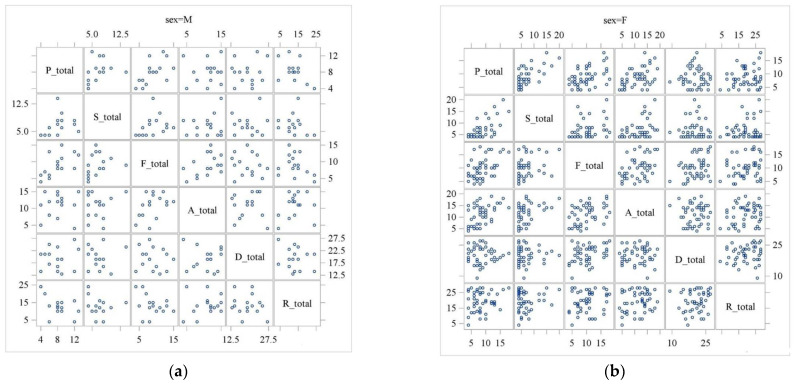
Scatter plot for combinations of men’s gaming (**a**) and women’s gaming (**b**) preference total values. A_total indicates Achievers total score. D_total indicates Disruptors total score. F_total indicates Free Spirits total score. P_total indicates Philanthropists total score. R_total indicates Players total score. S_total indicates Socializers total score.

**Figure 5 geriatrics-09-00021-f005:**
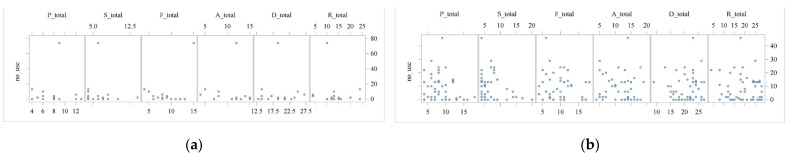
Scatter plot of the number of times the Choisoko system is used and the value for each game preference for men (**a**) and women (**b**). P-total indicates Philanthropists total score. S-total indicates Socializers total score. F-total indicates Free Spirits total score. A-total indicates Achievers total score. D-total indicates Disruptors total score. R-total indicates Players total score.

**Figure 6 geriatrics-09-00021-f006:**
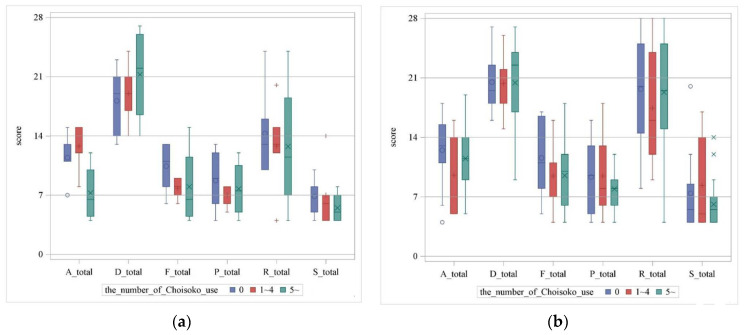
Boxplot for relationship with gaming preference value when the number of times the Choisoko system is used is divided iinto three groups for men (**a**) and women (**b**). Blue bar indicates number of times the Choisoko system had been used: 0 times. Red bar indicates number of times the Choisoko system had been used: 1 to 4 times. Green bar indicates number of times the Choisoko system had been used: 5 or more times.

**Figure 7 geriatrics-09-00021-f007:**
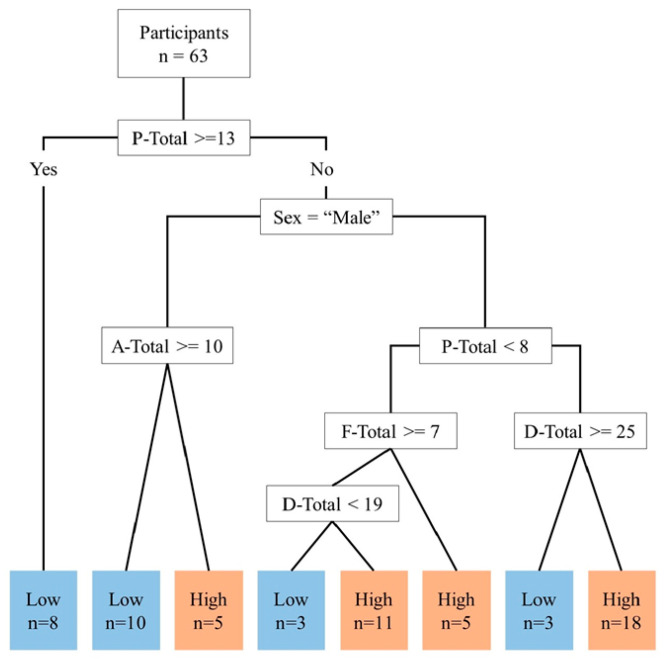
Analysis using decision tree regarding game preferences and number of times the Choisoko system is used. P-total indicates Philanthropists total score. A-total indicates Achievers total score. F-total indicates Free Spirits total score. D-total indicates Disruptors total score.

**Table 1 geriatrics-09-00021-t001:** Baseline characteristics.

Variable	Men Group		Women Group		*p*-Value
	N	Mean ± SD	N	Mean ± SD	
Age [year]	16	83.25 ± 3.02	47	82.13 ± 4.14	0.254
Height [cm]	16	161.5 ± 5.8	47	146.1 ± 5.8	<0.0001
Weight [kg]	16	60.4 ± 7.6	46	48.7 ± 7.1	<0.0001
BMI	16	23.1 ± 2.1	46	22.8 ± 3.3	0.718
Body fat percentage [%]	16	27.3 ± 7.1	44	33.0 ± 7.8	0.013
Alb [g/dL]	15	4.2 ± 0.3	47	4.2 ± 0.3	0.812
AST [U/L]	15	23.5 ± 7.7	47	26.7 ± 20.3	0.368
ALT [U/L]	15	17.3 ± 8.4	47	19.5 ± 23.8	0.593
BUN [mg/dL]	15	20.1 ± 6.1	47	18.2 ± 4.4	0.266
CRE [u/mL]	15	1.3 ± 0.9	47	0.8 ± 0.1	0.058
eGFR [mL/min/1.73m^2^]	15	51.5 ± 16.5	46	54.4 ± 10.5	0.532
WBC [/µL]	15	6420.0 ± 1286.3	47	6221.3 ± 1447.3	0.618
RBC [/µL]	15	448.5 ± 66.1	47	410.6 ± 47.5	0.054
Hb [g/dL]	15	13.8 ± 2.0	47	12.5 ± 1.4	0.034
Ht [%]	15	42.8 ± 5.6	47	38.7 ± 3.9	0.016
MCV [fL]	15	95.9 ± 3.9	47	94.7 ± 5.2	0.342
MCH [pg]	15	30.9 ± 1.9	47	30.6 ± 1.9	0.599
MCHC [%]	15	32.2 ± 1.3	47	32.3 ± 0.7	0.785
Platelet [10^3^/μL]	15	21.4 ± 3.8	47	23.2 ± 5.0	0.145
Grip strength (right) [kg]	16	31.2 ± 7.2	46	22.2 ± 3.8	<0.0001
Grip strength (left) [kg]	16	30.5 ± 7.0	46	21.2 ± 4.2	<0.0001
Five Times Sit-to-Stand Test [second]	15	9.5 ± 3.6	46	10.8 ± 5.3	0.293
Walk speed [km/h]	16	5.2 ± 1.1	46	6.6 ± 2.5	0.003
Walk speed max [km/h]	16	6.1 ± 8.6	46	5.4 ± 2.4	0.772
Muscle mass [kg]	16	23.6 ± 2.6	44	16.8 ± 2.0	<0.0001
Volume of iliopsoas muscle [mm]	15	52,986.7 ± 29,969.2	47	36,180.9 ± 16,571.1	0.054

SD: Standard deviation, CI: Confidence interval, Alb: albumin, AST: aspartate aminotransferase, ALT: alanine aminotransferase, BUN: blood urea nitrogen, CRE: creatinine, eGFR: estimated glomerular filtration rate, WBC: White blood cell, RBC: Red blood cell, Hb: hemoglobin, Ht: hematocrit, MCV: mean corpuscular volume, MCH: mean corpuscular hemoglobin, MCHC: mean corpuscular hemoglobin concentration.

**Table 2 geriatrics-09-00021-t002:** The average and median values of game preferences of the subjects by gender.

	Male (n = 16)		Female (n = 47)	
	Average (SD^※^)	Median (Range)	Average (SD^※^)	Median (Range)
P-total	7.94 (2.74)	8.0 (4–13)	8.57 (3.40)	8.0 (4–18)
S-total	6.56 (2.71)	6.0 (4–14)	6.87 (3.84)	5.0 (4–20)
F-total	9.99 (3.14)	8.5 (4–15)	10.02 (3.88)	10.0 (4–18)
A-total	10.81 (3.49)	11.5 (4–15)	11.38 (4.14)	12.0 (4–19)
D-total	19.19 (4.29)	19.0 (13–27)	20.43 (4.12)	21.0 (9–27)
R-total	13.44 (5.75)	13.0 (4–24)	19.04 (6.58)	19.0 (4–28)
age	83.25 (3.02)	82.5 (79–90)	82.13 (4.14)	82.0 (73–94)
No-use	7.31 (18.2)	2.0 (0–74)	9.02 (9.80)	6.0 (0–46)

SD^※^: Standard deviation No-use: Number of times of use of the choisoko system. P-total indicates Philanthropists total score. S-total indicates Socializers total score. F-total indicates Free Spirits total score. A-total indicates Achievers total score. D-total indicates Disruptors total score. R-total indicates Players total score.

**Table 3 geriatrics-09-00021-t003:** FIM scores after using the Choisoko system.

	Men Group		Women Group		
FIM Score	N	Mean ± SD	N	Mean ± SD	Mean Difference (95% CI)
Total score	10	115.20 ± 8.00	40	116.08 ± 3.41	0.88
					(CI: −4.90 to 6.65)
Exercise score	10	80.70 ± 6.46	40	81.35 ± 3.20	0.65
					(CI: −4.03 to 5.33)
Cognitive score	10	34.50 ± 1.58	40	34.73 ± 0.55	0.23
					(CI: −0.91 to 1.36)

Mean Differences: [the mean for women group] – [the mean for men group] FIM: Functional Independence Measure.

**Table 4 geriatrics-09-00021-t004:** Correlation coefficient between total game preference in male and female.

	Male							Female					
	P-Total	S-Total	F-Total	A-Total	D-Total	R-Total		P-Total	S-Total	F-Total	A-Total	D-Total	R-Total
P-total	1.00	0.36	0.60	0.12	0.11	−0.56		1.00	0.67	0.44	0.47	−0.08	0.16
S-total	0.36	1.00	0.27	0.15	0.09	−0.28		0.67	1.00	0.32	0.45	−0.02	0.10
F-total	0.60	0.27	1.00	0.43	−0.13	−0.32		0.44	0.32	1.00	0.41	0.31	0.21
A-total	0.12	0.15	0.43	1.00	−0.03	0.18		0.47	0.45	0.41	1.00	0.02	0.20
D-total	0.11	0.09	−0.13	−0.03	1.00	−0.21		−0.08	−0.02	0.31	0.02	1.00	0.16
R-total	−0.56	−0.28	−0.32	0.18	−0.21	1.00		0.16	0.10	0.21	0.20	0.16	1.00

P-total indicates Philanthropists total score. S-total indicates Socializers total score. F-total indicates Free Spirits total score. A-total indicates Achievers total score. D-total indicates Disruptors total score. R-total indicates Players total score.

**Table 5 geriatrics-09-00021-t005:** *T*-test results for male and female.

	Male				Female		
	No-Use† ≦ 1 (N = 7) Average (SD^※^)	No-Use† > 1 (N = 9) Average (SD^※^)	Difference in Mean Value (95% CI)		No-Use† ≦ 1 (N = 25) Average (SD^※^)	No-Use† > 1 (N = 22) Average (SD^※^)	Difference in Mean Value (95% CI)
P-total	8.71 (3.15)	7.33 (2.4)	1.38 (−1.59, 4.35)		9.20 (4.05)	7.86 (2.36)	1.34 (−0.65, 3.32)
S-total	6.86 (2.04)	6.33 (3.24)	0.52 (−2.49, 3.54)		7.76 (4.74)	5.86 (2.17)	1.90 (−0.32, 4.11)
F-total	10.43 (2.64)	7.89 (3.18)	2.54 (−0.66, 5.74)		10.16 (4.10)	9.86 (3.69)	0.30 (−2.01, 2.60)
A-total	11.43 (2.44)	10.33 (4.21)	1.10 (−2.76, 4.95)		11.24 (4.32)	11.55 (4.02)	−0.31 (−2.77, 2.16)
D-total	18.14 (3.85)	20.00 (4.66)	−1.86 (−6.54, 2.83)		20.00 (3.54)	20.91 (4.73)	−0.91 (−3.34, 1.53)
R-total	14.29 (4.86)	12.78 (6.57)	1.51 (−4.87, 7.88)		18.84 (6.46)	19.27 (6.87)	−0.43 (−4.35, 3.48)

SD^※^: Standard deviation, No-use†: Number of times of use of the choisoko system. P-total indicates Philanthropists total score. S-total indicates Socializers total score. F-total indicates Free Spirits total score. A-total indicates Achievers total score. D-total indicates Disruptors total score. R-total indicates Players total score.

**Table 6 geriatrics-09-00021-t006:** Spearman correlation coefficient between the number of times the Choisoko system used and the total value of each game preference.

	Male	Female
P-total	−0.22	−0.14
S-total	−0.33	−0.12
F-total	−0.37	−0.18
A-total	−0.26	−0.09
D-total	0.14	0.10
R-total	−0.16	−0.05

P-total indicates Philanthropists total score. S-total indicates Socializers total score. F-total indicates Free Spirits total score. A-total indicates Achievers total score. D-total indicates Disruptors total score. R-total indicates Players total score.

**Table 7 geriatrics-09-00021-t007:** Average and standard deviation of the number of times the Choisoko System used and each game preference for men and women.

	Men				Women		
	Use_c = 0 (N = 7)	Use_c = 1 (N = 5)	Use_c = 2 (N = 4)		Use_c = 0 (N = 12)	Use_c = 1 (N = 9)	Use_c = 2 (N = 26)
P-total	8.71 (3.15)	7.00 (1.41)	7.75 (3.50)		9.33 (4.36)	9.44 (4.69)	7.92 (2.21)
S-total	6.86 (2.04)	7.00 (4.12)	5.50 (1.91)		7.42 (4.7)	8.33 (5.39)	6.12 (2.58)
F-total	10.43 (2.64)	7.80 (1.3)	8.00 (4.97)		11.58 (4.19)	9.44 (4.22)	9.5 (3.56)
A-total	11.43 (2.44)	12.80 (2.95)	7.25 (3.59)		12.50 (4.19)	9.56 (4.77)	11.5 (3.84)
D-total	18.14 (3.85)	19.00 (3.81)	21.25 (5.91)		20.50 (3.45)	20.33 (3.64)	20.42 (4.66)
R-total	14.29 (4.86)	12.80 (5.81)	12.75 (8.38)		19.67 (6.27)	17.44 (7.16)	19.31 (6.7)

P-total indicates Philanthropists total score. S-total indicates Socializers total score. F-total indicates Free Spirits total score. A-total indicates Achievers total score. D-total indicates Disruptors total score. R-total indicates Players total score.

## Data Availability

In this study, the datasets generated and/or analyzed during the current study are available from the corresponding author upon reasonable request.

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
