# Peer review of "Synergistic Effect of Motivation for the Elderly and Support for Going Out II: Measures to Induce Elderly Men to Go Out"

_geriatrics, 2024, doi:10.3390/geriatrics9010021_

Round 1
Reviewer 1 Report
Comments and Suggestions for Authors
The authors examined an interesting question of how to motivate older people to go out more frequently. One previous study by this group showed that elderly women tend to go out more than elderly men. The authors hypothesized that elderly men might be encouraged to go out if offered to play games. The Choisoko system was used to call for transportation by people who required some level of assistance. Unfortunately, men remained less motivated to go out than women.
My enthusiasm about the manuscript was reduced due to the issues discussed below:
1. The introduction discussed the demographics in Japan in fine details, but lacked any background information on why men might be less interested in going out than women. While the idea that playing games could be more appealing for men than going shopping, the authors need to provide some scientific background for why this could be the case rather than relying on a single anecdotal evidence of one man asking to go to arcades. The introduction would also benefit from switching the focus from discussing the opportunities to go out to the motivation to go out.
2. The Choisoko system should be better described including how this system is different from any other system. It is unclear why calling to reserve transportation is better than, for example, using Uber (or other service alike).
3. Please explain how the 3:4 ratio of men to women is related to men's willingness to go out.
4. The methods section would benefit from a description of whether the participants had to pay for transportation called vie the Choisoko system.
5. It is unclear why the authors decided to collect muscle volume and blood indices. There was no justification for why these measures would affect the participants' willingness to go out. I am especially puzzled about the idea to collect the psoas muscle volume. Why was the preference given to that particular muscle rather than the glutes or quads?
6. Although the authors conducted a multitude of analyses, I don't think the results for male participants are reliable. There was only 16 men in the study. Some of the analyses included only a portion of this already small sample. I would suggest collecting more male participants.
7. There is no justification for the hypothesis that gaming preference is related to participants' motivation to go out. The paper would benefit from a more theory-driven approach.
Author Response
Authors’ responses to the Reviewers’ comments
The authors would like to thank the Reviewer for the constructive critique to improve the manuscript. We have made every effort to address the issues raised and to respond to all the comments. The revisions are indicated in red font in the revised manuscript. Please find below our detailed, point-by-point responses to your comments. We hope that our revisions have satisfactorily addressed your concerns.
For Reviewer 1
The authors examined an interesting question of how to motivate older people to go out more frequently. One previous study by this group showed that elderly women tend to go out more than elderly men. The authors hypothesized that elderly men might be encouraged to go out if offered to play games. The Choisoko system was used to call for transportation by people who required some level of assistance. Unfortunately, men remained less motivated to go out than women. My enthusiasm about the manuscript was reduced due to the issues discussed below:
1, The introduction discussed the demographics in Japan in fine details, but lacked any background information on why men might be less interested in going out than women. While the idea that playing games could be more appealing for men than going shopping, the authors need to provide some scientific background for why this could be the case rather than relying on a single anecdotal evidence of one man asking to go to arcades. The introduction would also benefit from switching the focus from discussing the opportunities to go out to the motivation to go out.
Response: We really appreciate your important suggestion. We agree with you. In the first demonstration experiment, it was difficult to even get elderly men to participate. Even after registering to participate, the Choisoko system was hardly ever used for elderly men. During the preparation period between the first and second demonstration experiment, we searched for factors that were overwhelmingly different between men's and women's preference and found that they had a taste for games. This is overwhelmingly higher for men. Additionally, as I mentioned in the main text, some of the men requested transportation to and from the game center. It is true that the first version omitted this process. Therefore, in the revised version, we have added the progress in the Introduction section from line 150 to 172.
2, The Choisoko system should be better described including how this system is different from any other system. It is unclear why calling to reserve transportation is better than, for example, using Uber (or other service alike).
Response: Thank you for your important suggestion. We have added details of the Choisoko system in the Introduction section from line 126 to 133.
3, Please explain how the 3:4 ratio of men to women is related to men's willingness to go out.
Response: Thank you for your important suggestion. The male to female ratio of 3:4 has no significant significance. So, we have changed the relevant part to ``Since there are approximately 15 million men over the age of 65,'' at line 43, 61, 153.
- The methods section would benefit from a description of whether the participants had to pay for transportation called vie the Choisoko system.
Response: We really appreciate your important suggestion. That's exactly right; support for going out should not place an economic burden on the elderly. In the appropriate parts, we have added “Consequently, the Choisoko system offers a cost-effective alternative to traditional taxis, mitigating financial strain on users. Many municipalities operate the Choisoko system, which collects a reasonable fare, but this demonstration experiment was conducted free of charge.” from line 130 to 133.
- It is unclear why the authors decided to collect muscle volume and blood indices. There was no justification for why these measures would affect the participants' willingness to go out. I am especially puzzled about the idea to collect the psoas muscle volume. Why was the preference given to that particular muscle rather than the glutes or quads?
Response: Thank you for your important suggestion. Since the subjects were elderly and this is a continuation of the first demonstration experiment, just in case, we measured laboratory parameters and muscle mass to avoid any problems during the demonstration experiment, and to elucidate the causes of changes in physical function. In addition, maintaining good posture is particularly important for elderly people, and decreased psoas muscle function may increase the risk of falls, so we measured the muscle mass. So, we have added above contents in 2.4 Laboratory and Muscle Mass Measurements from line 216 to 219 and from line 222 to 224.
6, Although the authors conducted a multitude of analyses, I don't think the results for male participants are reliable. There was only 16 men in the study. Some of the analyses included only a portion of this already small sample. I would suggest collecting more male participants.
Response: We really appreciate your critical suggestion. The biggest limitation in the Miyadera area, where we conducted 2nd demonstration experiments, was that no matter how hard we tried, we were only able to get 16 elderly men to participate. Therefore, our research team has five experts in statistical analysis, and we have taken this point into account and revised the Methods, Results, and Discussion sections to more accurately interpret the results. In addition, it is necessary to verify the results of this demonstration experiment through a large-scale survey, so we have added a note in the Conclusion section that we are planning to conduct a large-scale survey.
7, There is no justification for the hypothesis that gaming preference is related to participants' motivation to go out. The paper would benefit from a more theory-driven approach.
Response: We really appreciate your critical suggestion. As mentioned in the main text, there were no manuscripts regarding factors that induce elderly men to go out. Therefore, while preparing for the second demonstration experiment, we investigated the high preference that men have. One of them is gaming preferences. The main purpose of this research is to confirm whether gaming preferences and motivation for going out match. As a result, the present study showed some differences between elderly men and women among the six gaming preferences, and men with gaming preferences such as philanthropists, achievers, and free spirits were more likely to go out. However, only 16 elderly men participated in this demonstration experiment, which is the biggest limitation of this study. Therefore, future research will focus on older men's gaming preferences and dig deeper to uncover specific factors associated with this, which will include a large-scale survey.
Additional revision:
We have found numerical errors in Tables 5a and 5b. In the initial submission, 95% confidence intervals for differences in means were calculated using the Pooled method, which was changed to the Satterthwaite method to unify the methods from line 294 to 296. Also, the cutoff points for each group were changed to fit the rule based on medians from line 365 to 367.
Reviewer 2 Report
Comments and Suggestions for Authors
Please refer to the attached documents

Please refer to the attached documents
Author Response
Authors’ responses to the Reviewers’ comments
The authors would like to thank the Reviewer for the constructive critique to improve the manuscript. We have made every effort to address the issues raised and to respond to all the comments. The revisions are indicated in red font in the revised manuscript. Please find below our detailed, point-by-point responses to your comments. We hope that our revisions have satisfactorily addressed your concerns.
For Reviewer 2
This paper titled "Synergistic Effect of Motivation for the Elderly and Support for Going out II; Measures to Induce Elderly Men to Go Out" is interesting. Very good work, I suggest minor revision in some part in some sections in order to increase the quality and correctness of this article. The specific recommendations are as follows:
1, In Abstract, it is recommended to briefly describe the sample size and analysis method you choose.
Response: Thank you for your optimal suggestion. We have modified abstract section according to your points.
2, It is advisable to consider the 124th line as "A summary of the results is as follows. For elderly women, friends, shopping, convenience, and participation in events were considered to be factors with the potential to be effective motivating factors in guiding them to go out. They accepted this system, increased the number of times they went out, increased the number of walks, and showed a rehabilitation effect. On the other hand, men did not accept the system at all and did not respond at all to friends or shopping as factors that encouraged them to go out". This passage has been rephrased in the conclusion.
Response: Thank you for your substantial suggestion. The sentence you pointed out is in the Introduction section, and we have interpreted it as pointing out that its content overlaps with the beginning of the sentence in the Discussion section. So, we have modified applicable text in the Discussion section from line 439 to 441.
- In Table 3 of the 253rd line, the sample size of Men Group is N=10, does the Mean Difference test meet the necessary conditions in inferential statistics? In Table 5a of the 292nd line, the sample size of the groups are only 7 and 9, can the statistical testing analysis be performed?
Response: We really appreciate your critical suggestion. We agree with your essential comment. We have removed the p-values from Tables 3, 5a and 5b and changed the presentation of the results using point estimates and confidence intervals from line 331 to 336, line 371 to 375, 379 to 380 and 384 to 386. In addition, we have removed the description of the t-test in the methods section from line 268 to 272. So, we have changed to “2) Dividing the low- and high-frequency groups of the Choisoko system use, the frequency of use of the Choisoko system and six gaming preferences were compared. The low- and high-frequency groups were defined by the median for each sex.” from 272 to 274 and “All 95% confidence intervals (CIs) were two-sided, and all 95% CIs in mean differences were calculated using Satterthwaite method.” from line 293 to 296.
4, Please add some information in Figure 7. For example, consider the precision and accuracy of this decision tree.
Response: Thank you for pointing this out. The following text about the information in Figure 7 was added in the Results section. So, we have added “In the low user group, 72.4% (21/29) were correctly distinguished, and in the high user group, 91.2% (31/34) were correctly distinguished. Overall, 82.5% (52/63) were correctly classified by the tree.” from 417 to 421.
5, In section of conclusion, it is too condensed to suggest that the limitations of the study should be added (in addition to the exclusion of participants mentioned in lines 144 to 150) and more future studies (not limited to increasing the sample size).
Response: Thank you for your comment. As you indicated, we have revised the Conclusion section as following,
A second demonstration experiment was conducted with the main purpose of exploring factors that induce men to go out. The present study showed some differences between elderly men and women among the six gaming preferences, and men with gaming preferences such as philanthropists, achievers, and free spirits were more likely to go out. However, this study was limited in generalizability and precision: the limited geographic area and limited target population as described in Section 2.1, and small sample size. Therefore, more large-scale questionnaire surveys on various target populations are needed to confirm these findings, and they are currently in the planning stage.
Additional revision:
We have found numerical errors in Tables 5a and 5b. In the initial submission, 95% confidence intervals for differences in means were calculated using the Pooled method, which was changed to the Satterthwaite method to unify the methods from line 294 to 296. Also, the cutoff points for each group were changed to fit the rule based on medians from line 365 to 367.
Round 2
Reviewer 1 Report
Comments and Suggestions for Authors
Thank you for addressing my comments